# Inferring Inherent Optical Properties of Sea Ice Using 360-Degree Camera Radiance Measurements

Raphaël Larouche<sup>1,2,\*</sup>, Bastian Raulier<sup>1,3,\*</sup>, Christian Katlein<sup>4</sup>, Simon Lambert-Girard<sup>1</sup>, Simon Thibault<sup>2</sup>, Marcel Babin<sup>1</sup>

- <sup>1</sup>Takuvik International Research Laboratory, Université Laval (Canada) & CNRS (France), Département de Biologie and Québec-Océan, Université Laval, Pavillon Alexandre-Vachon 1045, avenue de la Médecine, Local 2064, G1V 0A6, Canada <sup>2</sup>Centre d'optique, photonique et laser (COPL), Université Laval, Québec, QC, Canada
  - <sup>3</sup>CERVO Brain Research Center, Québec, G1J 2G3, Canada
  - <sup>4</sup>Alfred-Wegener-Institut Helmholtz-Zentrum für Polar und Meeresforschung, Bremerhaven, Germany
- 10 \*These authors contributed equally to this work.

Correspondence to:

Raphaël Larouche (raphael.larouche@takuvik.ulaval.ca)

Bastian Raulier (bastian.raulier@takuvik.ulaval.ca)

Abstract. In this work, we demonstrate the utilization of a compact, consumer-grade 360-degree camera for measuring the in-ice spectral angular radiance distribution. This novel technique allows for the instantaneous acquisition of all radiometric quantities at a given depth with a non-intrusive probe. This gives the opportunity to monitor the light field structure (mean cosines) from the atmosphere to the underlying ocean beneath ice. In this study, we report vertical profiles of the light field geometric distribution measured at two sites representative of distinct ice types: High Arctic multi-year ice and Chaleur Bay (Quebec, Canada) landfast first-year ice. We also propose a technique to empirically retrieve the depth-resolved inherent optical properties by matching simulated profiles of spectral irradiances calculated with the HydroLight radiative transfer model to the observed ones. As reported in other studies, the derived reduced scattering coefficients were high (641.57 m<sup>-1</sup>, 72.85 m<sup>-1</sup>) in the top (2 cm, 5 cm) for both sites (High Arctic, Chaleur Bay) and lower in the interior part of the ice (0.48 to 4.10 m<sup>-1</sup>, 0.021 to 7.79 m<sup>-1</sup>). Due to the inherent underdetermined nature of the inversion problem, we emphasize the importance of using the similarity parameter that considers both the absorption and the reduced scattering coefficients. Finally, we believe that this radiometric device, combined with the proposed inversion technique, will allow to scale up the measurements of the inherent optical properties of different kinds of sea ice enabling to take better account of terrain variability in radiative transfer models.

#### 1 Introduction

The Arctic Ocean has undergone major transformations in the last few decades as perennial sea ice has largely been replaced by thinner first-year ice (Comiso, 2002; Maslanik et al., 2011; Stroeve et al., 2012; Tschudi et al., 2016) and as a significant decrease of the ice extent was observed (Comiso et al., 2008; Serreze et al., 2007; Stroeve and Notz, 2018). These changes have critical impacts on the atmosphere-ice-ocean system, especially during the spring to summer melting season as sea ice is transformed in a highly inhomogeneous cover of snow, bare ice, melt ponds and leads (Ehn et al., 2011; Frey et al., 2011;

Horvat et al., 2020; Katlein et al., 2016; Perovich et al., 2002). Seasonal ice has higher melt pond fraction (Eicken et al., 2004; Hunke et al., 2013; Li et al., 2020) and enables more solar shortwave radiation transmission and heat deposition through absorption (Nicolaus et al., 2012; Stroeve and Notz, 2018). This new energy partition warms sea ice and the underlying ocean leading to accelerated ice melt through a positive feedback loop (Arndt and Nicolaus, 2014; Curry et al., 1995; Perovich et al., 2008) which impacts regional Arctic climates. Thinner ice cover promotes significant under-ice phytoplankton blooms earlier in the season (Arrigo et al., 2012, 2014; Mundy et al., 2009).

40

The amount of shortwave radiation scattered or absorbed by a given medium is determined by its inherent optical properties (IOPs). These material properties determine the propagation of light throughout the medium, which in bulk are often described by the apparent optical properties (AOPs) like albedo and transmittance (Light et al., 2003). The apparent optical properties of sea ice exhibit large seasonal and spatial variability due to the mosaic-like surface structure and variations in the physical properties of sea ice (Katlein et al., 2019; Matthes et al., 2020; Perovich et al., 1998). An improved understanding of the links between structural and optical properties is needed to predict the impacts of sea ice transformation on Arctic ecosystems. While surface AOPs such as albedo (Ehn et al., 2006; Grenfell and Perovich, 2004; Perovich et al., 2002) and bulk transmittance (Katlein et al., 2015; Nicolaus and Katlein, 2013; Perovich et al., 1993, 1998) have been well documented over the past decades, very few in situ optical measurements have been made inside sea ice. These additional measurements are needed to better constrain optical models. Vertical profiles of planar irradiances have been acquired with large probes—which are prone to self-shadowing—lowered into bore holes of 5 to 10 cm in diameter (Ehn et al., 2008b; Light et al., 2008, 2015; Xu et al., 2012a) or, alternatively, measured by digging holes from below the sea ice which required a diver (Ehn et al., 2008a). Efforts to reduce perturbations on sea ice physical properties were made with the development of vertical arrangements of fiber optics (Wang et al., 2014) or photodiodes having their normal axis 90° rotated from zenith direction (Katlein et al., 2021) taking autonomous measurements of irradiance in refrozen holes. Acquiring either planar or scalar irradiance is not sufficient to fully describe the internal light field because both quantities average the angular distribution of radiance. These measurements lack information about the light field geometry within the ice. Measurements of radiance have previously been collected in sea ice using single-direction radiance meters with a 3° to 7° field of view (FOV) (Pegau and Zaneveld, 2000; Perovich et al., 1998; Xu et al., 2012a). This approach is, however, time consuming as it requires drilling holes in different zenith directions and rotating the radiometer for azimuth measurements. This process results in full radiance distribution but suffer from low angular resolution and high structural disturbance by the sampling method.

We present a solution to measure radiance using a camera assembly whose pixels instantaneously resolve angular radiance distributions over a large portion of  $4\pi$  steradians. This enables the measurement of all radiometric quantities and, consequently, all AOPs that can be calculated from angular radiance distribution (Mobley, 1994). Such instruments have been used to study radiative transfer in the ocean (Antoine et al., 2013; Smith et al., 1970; Voss and Chapin, 1992, 2005; Voss and Zibordi, 1989; Wei et al., 2012). The smallest of these radiometers, which measured radiance, was packaged inside a 9.6 cm

x 26 cm (diameter x length) case (Antoine et al., 2013), making it unsuitable for in-ice applications due to self-shadowing and the large hole that must be drilled. Recently, lens and imaging sensor miniaturization has led to the commercialization of compact fisheye cameras. One of them is the Insta360 ONE (see Fig. 1c. Insta360, Arashi Vision Inc.). After rigorous radiometric calibrations (Larouche et al., 2024b) this low-cost camera becomes an easy-to-use instrument for measuring radiance distributions within sea ice. This paper presents the first vertical profiles of the angular radiance distributions inside sea ice at high angular resolution. The measurements were acquired in two different field sites, one in the High Arctic close to the geographic North Pole and the other one in Chaleur Bay in the province of Quebec. First, we provide the vertical profile of the radiometric quantities for High Arctic multi-year sea ice and then some depth resolved AOPs. For both field sites, we present IOPs inferred from the radiometric quantities calculated with the HydroLight (HL) radiative transfer simulation software that best fit the observed ones.

# 2 Materials and methods

# 2.1 Field measurements

High-Arctic field measurements were made near the geographic North Pole on August 31, 2018 (89° 25.21'N, 63° 08.67'E) during the AO18 expedition with the research icebreaker *Oden* (see Fig. 1a and 1d). The multi-year (MY) sea ice at the site was 185 cm thick with a freeboard of 17 cm and covered by 2 cm of fresh dry snow. Profiles of angular radiance distributions were collected inside a 5 cm diameter hole, snugly fitting the camera case. Measurements were made at a vertical resolution of 20 cm. At the measurements site, three sensors RAMSES-ACC-VIS (TriOS GmbH, Rastede, Germany) were positioned to acquire the downward, reflected, and transmitted (under sea ice) planar irradiance (Fig. 1c). Spectral irradiance was measured from 320 to 950 nm with increments of 3.3 nm (Katlein et al., 2021; Nicolaus et al., 2010). As for the atmospheric conditions, the incoming radiation was highly diffuse (see Fig. 1a) because of the presence of low stratus clouds (often observed in the High Arctic) and the low sun elevation.

A second field campaign took place in Chaleur Bay, Quebec, Canada (48° 06.47' N, 66° 26.97' W) on March 23, 2022 (see Fig. 1b). This place is located in the Gulf of St. Lawrence where seasonal sea ice forms during winter. The sampling site was chosen close to land. Four different holes were drilled at the site. The thickness of the ice varied from 72 cm to 80 cm, while freeboard was within 15 cm to 20 cm. This unusually high freeboard is explained by the landfast ice attached to the coast, even at low tide. The surface of the ice was covered with very granular snow, indistinguishable from sun-transformed ice. To underline this ambiguity, this part of the ice is referred as "surface slab". Vertical profiles of angular radiance distribution were acquired in each hole with depth increments of 5 cm. The Compact-Optical Profiling System (C-OPS, Biospherical Instruments Inc.) was used to continuously collect the downwelling irradiance (19 spectral bands between 380 and 875 nm; (Morrow et al., 2010)) as surface reference during in-ice camera measurements. As seen from Fig. 1b, clear sky conditions prevailed at the sampling site with a few passing clouds. The sun elevation at the time of measurements was between 40-43°. At both field

locations, the experimenters paid particular attention to site selection in order to ensure the largest possible homogeneous area.
No keels or under-ice variability were observed. Each profile measurement took approximately 10 to 15 minutes, depending on the vertical resolution and ice thickness.

Figure 1: Field sampling of the depth-resolved angular radiance distribution inside sea ice using the commercial 360-degrees camera Insta360 ONE. In (a), the image taken with the omnidirectional camera at the site near the North Pole in High Arctic (89° 25.21'N, 63° 08.67'E) shows a sun near the horizon and a mostly overcast sky, which are conditions producing an incident light field close to isotropic. (b) Sampling site at Chaleur Bay (48° 06.47'N, 66° 26.97'W), Quebec, Canada, with completely different meteorological conditions (sunny day with very few clouds) where one can see the operator inserting inside sea ice the graduated stick with the camera attached to the tip. Schematic of the acquisition method is illustrated in (c) with the camera inserted in the drilled hole (5 cm in diameter) for image capture at multiple depths. In High Arctic, RAMSES-ACC-VIS sensors were used to measure irradiance at the surface and below sea ice, while a C-OPS irradiance sensor was used (at surface only) at Chaleur Bay. (d) Structural model associated with the different ice layers used for inversion of optical properties.


# 2.2 Camera description and capture modes





The Insta360 ONE low-cost omnidirectional camera (see Fig. 1c) has a diameter of 5 cm, includes two fixed-aperture (f#=2.2) fisheye lenses. The imaging detectors are two Sony CMOS sensors (1/2.3" format, 6.95 mm in diagonal) with a total number of 3456 x 3456 pixels (12 mega-pixel) covered in a repeated Bayer mosaic of 3 waveband filters (conventional RGB). The analog to digital converter (ADC) has a resolution of 14 bits (16 384 possible values). When the camera is in air, each imaging sensor can capture light from a hemispherical solid angle of  $2\pi$  steradians. In water, this solid angle is reduced due to a decrease in the field of view along the optical axis of each lens, from 90° to 76° (Larouche et al., 2024b). In our measurement set-up, the optical axes (aligned with the zc axis in the zoomed region of Fig. 1c) of both fisheye lenses are oriented 90° from the zenith. An important feature of this camera is the availability of raw image capture at sensor-level allowing radiometry utilization. The camera can be purchased with a waterproof enclosure, and thus can be used in wet environments such as holes drilled in sea ice. With the camera attached to a depth-graduated stick, the acquisition strategy was to start a timer – set to the maximum value of 10 s – and then quickly lower the camera to the desired location inside the drill hole (see Fig. 1c). With the raw option activated, the images were saved to a microSD card in Digital Negative format (DNG, developed by Adobe).

## 2.3 Image processing, radiometry, and optical properties

The image processing pipeline starts by performing demosaicing, where each spectral RGB component is downsampled from the raw image. Then, after dark correction, we apply the measurement equation with the proper calibration parameters – considering in-air/in-water utilization – to transform the digital numbers of each spectral i band into effective spectral radiance values  $\bar{L}_i$  [W sr<sup>-1</sup> m<sup>-2</sup> nm<sup>-1</sup>]. For the dark subtraction, an average of the unexposed part of the CMOS sensor was used. The measurement equation as well as the calibration and characterization methodologies of the variables involved in this formula are fully described in (Larouche et al., 2024b). Lastly, the pixel wise  $\bar{L}_i(\theta_c, \phi_c)$  – with  $\theta_c, \phi_c$  being the spherical coordinates with respect to the optics reference systems – are re-mapped on a zenith ( $\theta$ ) and azimuth ( $\phi$ ) grid of 1° in angular resolution.

The angular distribution of radiance carries a large amount of information; the integration of the radiant energy in every direction (over a hemispherical solid angle) weighted by  $\cos \theta$  gives the equivalent planar irradiance. This radiometric quantity is used for energy budget calculation influencing sea ice mass balance (Ebert et al., 1995; Jin et al., 1994; Light et al., 2015). Scalar irradiance is a more appropriate radiometric quantity for photosynthetically active organisms as they are equally sensitive to every photon direction (Morel, 1991). Both quantities can be calculated given:

$$E(z,\lambda) = \int_{0}^{2\pi} \int_{\theta_{1}}^{\theta_{2}} f(z,\theta,\phi) d\theta d\phi \tag{1}$$

with  $E(z,\lambda)$  [W m<sup>-2</sup> nm<sup>-1</sup>] being the irradiances at different depths z [cm]. For a planar irradiance of  $(E_d \text{ or } E_u)$ ,  $f(z,\theta,\phi) = \bar{L}_i(z,\theta,\phi) |\cos\theta| \sin\theta$ . With a scalar irradiance of  $(E_d^o \text{ or } E_u^o)$ , the integrand function is  $f(z,\theta,\phi) = \bar{L}_i(z,\theta,\phi) \sin\theta$ . For the

radiation coming from either downward or upward directions (d and u subscripts), the zenithal integration boundaries are set to  $[\theta_1, \theta_2] = [0, \pi/2]$  and  $[\theta_1, \theta_2] = [\pi/2, \pi]$ , respectively. We used the composite Simpson's rule for the numerical integrations in both dimensions ( $\theta$ ,  $\phi$ ). However, before calculating the irradiances, any missing radiance values—resulting from the in-water reduced FOV—had to be extrapolated. This extrapolation was accomplished through a 5<sup>th</sup> degree Legendre polynomial fit on the azimuthally averaged spectral radiance (see appendix A). The attenuation with depth of the planar irradiance travelling in the downward hemisphere is calculated as:

$$K_d(z',\lambda) = -\frac{2}{E_d(z_{k+1}) + E_d(z_k)} \cdot \frac{E_d(z_{k+1}) - E_d(z_k)}{z_{k+1} - z_k} \tag{2}$$

where  $K_d(z',\lambda)$  [m<sup>-1</sup>] are the downwelling diffuse attenuation coefficients at the central depths  $z'=0.5 \cdot (z_{k+1}+z_k)$ , midway between two consecutive discrete measurements  $z_k$  and  $z_{k+1}$  [m]. This function gives the rate at which light gets attenuated due to scattering and absorption. In addition to being determined by the IOPs, the  $K_d(z',\lambda)$  profile is also influenced by structure of the ambient light field determined by the environmental conditions such as the position of the sun in the sky. Therefore, it is an apparent optical property.



The average cosine links together the planar and scalar irradiances for the downward and upward light field according to:

$$\bar{\mu}_d(z,\lambda) = \frac{E_d(z,\lambda)}{E_d^0(z,\lambda)} \qquad \bar{\mu}_u = \frac{E_u(z,\lambda)}{E_u^0(z,\lambda)} \tag{3}$$

where  $\bar{\mu}_d$  and  $\bar{\mu}_u$  are respectively the average cosines for downwelling and upwelling light fields. They are geometric indices of the radiance angular distribution and their evolution is driven by the medium inherent optical properties. For  $\mu_d$ , values of 0.5 and 1.0 correspond to isotropic and completely downward light fluxes, respectively. The average cosine of the complete angular radiance distribution is calculated from the following relationship:

$$\bar{\mu}(z,\lambda) = \langle \cos \theta \rangle = \frac{\int_0^{2\pi} \int_0^{\pi} \bar{L}_i(z,\theta,\phi) \cdot \cos \theta \cdot \sin \theta \, d\theta d\phi}{\int_0^{2\pi} \int_0^{\pi} \bar{L}_i(z,\theta,\phi) \cdot \sin \theta \, d\theta d\phi} = \frac{E_d(z,\lambda) - E_u(z,\lambda)}{E_o(z,\lambda)} \tag{4}$$

where  $\bar{\mu}$  and  $E_o$  [W m<sup>-2</sup> nm<sup>-1</sup>] are respectively the average cosine and the scalar irradiance of the  $4\pi$  steradians sphere. Apart from understanding how the geometry of the light field evolves with depth, paring the average cosine  $\bar{\mu}(z,\lambda)$  with the net irradiance  $E_{net}(z,\lambda) = E_d(z,\lambda) - E_u(z,\lambda)$ , gives an estimation of the depth resolved absorption coefficient  $a(z,\lambda)$  [m<sup>-1</sup>] under conservation of energy (Mobley, 1994):

$$a(z',\lambda) = -\frac{\bar{\mu}(z_{k+1}) + \bar{\mu}(z_k)}{2} \cdot \frac{\ln[E_{net}(z_{k+1},\lambda)/E_{net}(z_k,\lambda)]}{z_{k+1} - z_k}$$
(5)

This equation, also known as Gershun's law, resolves the transport equation explicitly assuming negligible contributions from inelastic scattering or internal sources. It has been used to infer absorption coefficients in natural waters with an uncertainty in the range of 21 % (Voss, 1989).

The other inherent optical properties, the scattering coefficient b(z) [m<sup>-1</sup>] and the phase function  $p(\theta)$  [sr<sup>-1</sup>], cannot be calculated explicitly. However, they can be estimated by carefully matching radiative transfer simulations outputs to the measured radiometric quantities or apparent optical properties (Light et al., 2008; Mobley et al., 1998; Perovich, 1990; Xu et al., 2012b). The phase function describes the probability density that a photon gets redirected at a certain angle after a single scattering event. In sea ice (assuming negligible Rayleigh scattering) the scattering events occur mainly after a photon reaches an interface between ice and air or brine. This scattering probability in sea ice is assumed spectrally invariant, a correct simplification in Mie regime when the scatterers are large and distant as diffraction and interference effects are negligible (Grenfell, 1983). For radiative transfer simulations in sea ice, the one-term Henyey-Greenstein (HG) function (Henyey and Greenstein, 1941) is usually adopted to approximate the phase function (Light et al., 2003, 2004, 2008, 2015; Mobley et al., 1998; Petrich et al., 2012):

$$p_{HG}(\theta, g_{HG}) = \frac{1}{4\pi} \cdot \frac{1 - g_{HG}^2}{(1 + g_{HG}^2 - 2g_{HG}\cos\theta)^{\frac{3}{2}}}.$$
 (6)

The asymmetry parameter (anisotropy coefficient) q varies between 0 (isotropic) and 1 (complete forward) and is equal to:

$$g = \langle \cos \theta \rangle = 2\pi \int_{0}^{\pi} p(\theta) \cos \theta \sin \theta d\theta . \tag{7}$$

The one-term Henyey-Greenstein equation was proposed from observations of the scattering by interstellar matter and has only one degree of freedom. It fails to accurately model complex phase functions; the latter being physically determined by the shape, the size distributions, and the complex refractive index of the scatterers. In sea ice where most of the layers have large scattering coefficients, the diffusion regime is often reached so that the detailed angular shape of the phase function becomes irrelevant and only its first moment – driving the front-to-back scattering ratio – has relevance for the radiative transfer (Jacques, 2013). The radiative transfer simulations presented in this work therefore uses the Henyey-Greenstein one-term equation.

#### 2.4 Inherent optical properties inversion





There is a wide variety of models used to solve the radiative transfer equation (RTE) (Preisendorfer, 1965) in scattering media. In this study we chose the HydroLight (Mobley, 1994) radiative transfer model which solves the RTE using the invariant embedding technique. This model takes as input any depth dependant set of IOPs and sky incident irradiance.

The two measurement sites were treated differently in HydroLight because they differ in many respects. For the High Arctic, we set a diffuse sky illumination to simulate the stratus cloud cover that day (see. Fig. 1a). Several homogeneous layers were used for the ice structural model (see Fig. 1d). At the surface, a 2 cm snow layer on top of ice was set, followed by a layer from snow to water level. Interior ice was divided in slabs of 40 cm, except for the first layer below water level set to 37 cm, to include at least one point of comparison with field measurements. The eight remaining centimeters above the ice-seawater

interface were considered the skeletal layer. In the High Arctic, the sea floor reflectance effect was neglected, the water column being simulated as infinitely deep. For the measurements taken under clear sky in Chaleur Bay, the geographic coordinates of the sampling site and the UTC time of measurement were used to calculate the proper zenithal angle of the sun within the model. For this site, we considered a 5 cm surface layer also followed by a layer down to freeboard level. Inner ice layers were then determined by inspecting the profile of absorption coefficients calculated using Gershun's law and that of the diffuse attenuation coefficient. Then, to properly represent the shallow waters of the bay, the simulations were run with a water column beneath the ice of 5 meters with a sea floor reflectance of 10 %. This reflectance value is representative of coastal oceanic bottom reflectance of intertidal ecosystems near Chaleur Bay (Légaré et al., 2022). For both measurement sites, the downwelling spectral irradiance measurement at the surface was used as input in the simulations, while the shape of the radiance was calculated with Hydrolight using atmospheric variables adapted to recreate the conditions observed in the field. The changes in refractive indexes in the snow-ice-ocean system were not considered (Ehn et al., 2008a, b; Jin et al., 1994). This simpler treatment avoids the problem of enhance downward irradiance (EDI) at the atmosphere-sea ice interface (Jiang et al., 2005). It also helps removing the ambiguity of choosing the position of the air-ice interface when there is a snow layer on top. These interfaces are simply treated as additional scattering events in our simulations. We acknowledge that these are assumptions and that more efforts would be required solely on understanding how to properly model the interfaces and their 210 refractive index in sea ice.

To constrain the optical properties, we decided to fix the absorption coefficients. In the case of the High Arctic, it is reasonable to assume that the ice itself is responsible for most of the absorption. This assumption was also made in the inversions reported by previous studies (Light et al., 2008, 2015). We used the absorption spectral coefficients reported by Grenfell and Perovich (1981). In the case of Chaleur Bay, we observed the presence of impurities in sea ice, likely microalgae on-site. We therefore used the absorption coefficients calculated by Gershun's law (Eq. (5)), taking the median of the values inside each layer. To further constrain the problem, we also fixed the g parameter of the phase function. Below the surface layer, the HG function was given a constant high asymmetry parameter of 0.99 (highly forward peak scattering) that is in accordance with previous works on radiative transfer inside sea ice (Ehn et al., 2008b; Light et al., 2004; Maffione et al., 1998; Mobley et al., 1998; Petrich et al., 2012). Near the surface, we set g to 0.85 based on literature (Light et al., 2004) as previous Mie calculations give an asymmetry parameter in that vicinity from larger volume fraction of air bubbles in the laver.

To infer the scattering coefficient profile of the two sites, we developed a recursive inversion algorithm using the Nelder-Mead downhill simplex algorithm (Nelder and Mead, 1965). The algorithm goes as follows and repeated N times until satisfactory results are obtained.

#### 1. Repeat N times:






For each layer in the profile:

- i. Run the Nelder-Mead algorithm varying the scattering coefficient until the layer's measured and simulated reflectance are within 1 % or reaching the maximum iteration number *M*.
- 2. Run the Nelder-Mead algorithm varying the entire profile of scattering coefficients, stops when loss  $\mathcal{L} \leq 0.01$  (see Eq. (8)) or reaching the maximum iteration number G.

This algorithm comprises two distinct steps. Firstly, it recursively uses each layer discrete reflectance as an indicator, this apparent optical property approaches an intrinsic characteristic of the layer and is little influenced by the others. This reduces the dimensionality of the problem in several small succinct steps, enabling the Nelder-Mead method to converge more rapidly. With the first stage having converged to the vicinity of the global minimum, the second stage proceeds to the minimization over all irradiance profiles. The loss  $\mathcal{L}$  [-] used at that second step is the sum of the relative error for the different irradiances  $(E_d, E_u, \text{ and } E_o)$  for each spectral band i and at discrete measurement depth  $z_k$  [cm]:



$$\mathcal{L} = \sum_{i=0}^{I} \sum_{k=0}^{K} \frac{\left| E_{d,i,HL}(z_k) - E_{d,i,Field}(z_k) \right|}{E_{d,i,HL}(z_k)} + \frac{\left| E_{u,i,HL}(z_k) - E_{u,i,Field}(z_k) \right|}{E_{u,i,HL}(z_k)} + \frac{\left| E_{o,i,HL}(z_k) - E_{o,i,Field}(z_k) \right|}{E_{o,i,HL}(z_k)}$$
(8)

where the subscripts *HL* and *Field* refer to the simulation and the measurement respectively. This way of calculating the error enables fine-tuning of the inverted values to obtain the simulated irradiance profile closest to the measured one.

This procedure, although leading to profiles very close to those observed, does not necessarily recover the actual inherent optical properties of the ice. This procedure only allows finding one of the solutions among a larger ensemble. This ensemble is defined as the sets of inherent optical properties leading to the same irradiance gradient. For example, we can imagine two different media: one very absorbent but with low scattering and another with little absorption but high scattering. Both would result in the same irradiance gradient. Van de Hulst introduced in 1980 a useful equation, Eq. (9), to compare the triplet of IOPs that give rise to the same gradient [68].

$$S = \sqrt{(1 - \omega_o)/(1 - \omega_o g)} = [1 + (b/a) \cdot (1 - g)]^{-1/2}$$
(9)

Where S is the similarity parameter and  $\omega_o = b/(a+b)$  is the single scattering albedo. This invariant is the geometric mean of the two first moments of the diffusion pattern. The latter being valid for thick scattering slabs far from boundaries, it has already been used to compare set of IOPs by Light and Ehn (Ehn et al., 2008b; Light et al., 2004). Recalling Eq. (9) with the reduced scattering coefficient, if the ratio  $b'/a \gg 1$  (mainly scattering medium), then the similarity parameter will tend toward 0. Conversely, a ratio  $b'/a \ll 1$  (mainly absorbing medium) means that the S will get closer to unity. Also, one medium that scatters isotropically may be equivalent radiometrically to another one with a higher scattering coefficient and an asymmetry parameter closer to unity. This gives rise to:

$$b' = b(1 - g) \tag{10}$$

with b' corresponding to the reduced scattering coefficient, another similarity parameter. This quantity is often used for comparisons of inverted IOPs in sea ice as two regions with the same reduced scattering coefficients are identical in term of scattering.

#### 2.5 Inversion uncertainties

The inversion described in the previous section certainly leads to errors when inferring inherent optical parameters. Therefore, we developed an experiment to estimate the error level that percolates through the process. For that purpose, we used HydroLight to generate the radiometric quantities of a set of fifty different ice conditions following the structural model proposed in previous inversions (Light et al., 2008, 2015). In each case, the ice was 150 cm thick and consisted of a surface scattering layer (5 cm), a drained (27.5 cm) and interior layer (117.5 cm). The conditions differed by their reduced scattering coefficient profile, for each of the layers a random value was drawn from uniform distributions. The boundaries of these distributions were b' ∈ [20, 150] m<sup>-1</sup> (g = 0.85) for the surface scattering layer, [2.4, 12] m<sup>-1</sup> (g = 0.99) for the drained layer, [0.5, 1.8] m<sup>-1</sup> (g = 0.99) for the interior layer and [0.1, 1.0] m<sup>-1</sup> (g = 0.90) for the seawater. These values correspond to the expected values for the various layers for first-year ice (Light et al., 2008, 2015). Then, the inversion algorithm was used to infer the random drawn scattering coefficients for all the layers. To increase the complexity of the problem, the starting points for the inference were also sampled randomly. In this way, it was possible to check whether the distance between the desired start and end points had any impact on the error.

# 3 Results

# 3.1 Field measurements

# 3.1.1 High Arctic

Figure 2 shows the spectral angular radiance distributions as measured by the camera for the three spectral bands. In the first row (Fig. 2a-c), the spectral components of the geometric light field L̄(θ, φ) for a depth of 40 cm are shown while the second row (Fig. 2d-f) displays normalized-radiance distributions vertically stacked for depths between 40-160 cm with increments of 40 cm. These polar graphs show radiance angular distribution in spherical coordinates. The azimuth angle corresponds to a fixed reference on the camera, with lens #1 arbitrarily set at 90 degrees and lens #2 at 270 degrees when transforming RAW images into radiance. However, as the radiative field in sea ice is considered to be homogeneous as a function of azimuth angle, the camera is not positioned in exactly the same way for each measurement. This could have an impact in ice with very low scattering or under a melt pool, but not in the two cases studied. The zenith angle indicates where radiance comes from relative to the vertical axis. It varies from 0 (center of graph, downward direction) to 160 degrees (outer ring, upward direction) and indicates the elevation of the energy direction where 0 degrees indicates a downward direction (towards the ocean) and
180 degrees would indicate a perfectly upward direction (towards the atmosphere). The top panels (a, b and c) show radiance

with the same color scale for the three channels centered at 480, 540 and 600 nm. The signal is predominantly blue in a downward direction (center of graph), followed by green and red. In the bottom panels (d, e and f), radiance is normalized to each depth, allowing us to better appreciate how its shape changes with depth. At higher elevations, the signal is much more homogeneous, whereas deeper within the sea ice, the angular distribution of radiance becomes increasingly downward. This effect is further accentuated at longer wavelengths. Because the lost (from top to bottom) of the radiation travelling upward ( $\theta > 90^{\circ}$ ) is larger compared to the removal of photons below 90°. From 60 cm to 200 cm,  $\bar{\mu}_d$  varies between 0.521-0.555 at 540 nm and between 0.531-0.562 at 600 nm (see full curves of Fig. 3d). For the upwelling light field,  $\bar{\mu}_u$  varies from 0.483-0.336 (540 nm) and 0.482-0.333 (600 nm) for the same depth interval (60-200 cm). As an average cosine for the upwelling photons of 1.0 represents completely straight-upward light fluxes ( $\theta = 180^{\circ}$ ), reduction of  $\bar{\mu}_u$  with depth suggests that we gradually find less radiation at larger zenith angles. Another apparent optical property, the diffuse attenuation coefficient, gives insights on the presence of different layers in the ice column. Figure 3h shows these coefficients for the downward light field. The bulk diffuse attenuation coefficients for the total ice thickness are 1.35 m<sup>-1</sup> (480 nm), 1.37 m<sup>-1</sup> (540 nm), and 1.58 m<sup>-1</sup> (600 nm). At the top (0-20 cm), the average  $K_d$  are 4.24 m<sup>-1</sup> (480 nm), 4.39 m<sup>-1</sup> (540 nm), and 4.84 m<sup>-1</sup> (600 nm). Just below, an intermediate layer sits from 20 to 80 cm, with average values of 1.69 m<sup>-1</sup>, 1.75 m<sup>-1</sup>, and 2.08 m<sup>-1</sup> for the blue, green, and red spectral bands, while between 80-180 cm, these  $K_d$  decrease to averages of 0.56 m<sup>-1</sup>, 0.54 m<sup>-1</sup>, and 0.63 m<sup>-1</sup> (see Fig. 3h).

Figure 2: Angular spectral radiance distribution [W sr<sup>-1</sup> m<sup>-2</sup> nm<sup>-1</sup>] as measured by the camera in High Arctic (AO2018 expedition). The top row (a), (b) and (c), displays the light fields at 40 cm depth for the blue (480 nm), green (540 nm), and red (600 nm) bands respectively. The zenith coordinates  $\theta$  corresponds to the radial circular lines while the azimuth  $\varphi$  are the angular lines. The white regions are the missing values over the  $4\pi$  steradians sphere due to the field of view reduced to 76° in water. The second row presents the radiance distributions at various depths (40, 80, 120 and 160 cm), but normalized to their respective maximum. The blue (480 nm), green (540 nm), and red (600 nm) bands are shown respectively in (d), (e), and (f).

Figure 3a-c present the planar downward, planar upward and total scalar irradiances used for IOPs inversion. We observe that the simulated irradiances closely align with the measured ones. Notably, the average relative differences are 10.3, 7.2, and 6.3 % for the planar downwelling, planar upwelling and scalar total light fields, respectively. The relative differences spectrally averaged are all below 15 % at depths between 20-200 cm for  $E_d$ ,  $E_u$  and  $E_o$ . Indeed, higher discrepancies are found as we get closer to the upper and lower boundaries. This is particularly obvious when looking at average cosines  $\leq$  20.0 cm for the upwelling (Fig. 3e), and total (Fig. 3f) radiance distributions, where surface hole effects seem to have perturbed the measurements (displayed as full lines). Table 1 shows the set of IOPs profiles associated with the modeled irradiances. The absorption coefficient used in the simulations (full line) and the one calculated with the Gershun's law (Eq. (5) are displayed in Fig. 3i. The a(z) isolated from Gershun's law are not constant as a function of depth, unlike the pure ice values used for

HydroLight simulations at that site. At the top, this may be due to surface effects that contaminated our measurements. Some of the Gershun's law-inferred absorption coefficients are negatives (80-100 cm). These are caused by the increase of the net irradiances between the two measurements at 80 and 100 cm. (see Fig. 3g). We decided not to show them in Fig. 3i as they likely derive from large observational uncertainties. Nonetheless, we interestingly noticed – by taking the depth median of all the Gershun's law in-ice values – that the absorption coefficients of 0.061, 0.054, and 0.138 m<sup>-1</sup> for the blue, green, and red channels are very close to those for pure bubble free ice of 0.043, 0.0683, 0.12 m<sup>-1</sup> (Grenfell and Perovich, 1981). Table 1 gives the fitted scattering properties, where we clearly observe distinctions between the layers. At the surface, the modeled snow layer has the largest b' of 641.57 m<sup>-1</sup>. Just below, the reduced scattering coefficient drops significantly to 4.10 m<sup>-1</sup> and remains roughly constant in the two subsequent layers with b' being 3.96 and 2.92 m<sup>-1</sup> (20-57 and 57-97 cm). The inferred coefficient then decreases to 0.98 and 0.48 m<sup>-1</sup> (97-137 and 137-177 cm). Lastly, we note an increase in scattering in the last layer to 2.77 m<sup>-1</sup> (177-185 cm).

Table 1. Inherent optical properties (IOPs) inverted from HydroLight and depth-resolved angular radiance distributions measured in High Arctic (AO2018 expedition). The absorption coefficients a [m<sup>-1</sup>], the scattering coefficients b [m<sup>-1</sup>], the anisotropy coefficient of the phase function g, the reduced scattering coefficient b(1-g) are given for a ice slab splitted into one drained layer (DL) above the freeboard and five layers below it. The layers are made of old interior ice (OII), young interior ice (YII) and skeletal layer (SL). The dimensionless similarity parameters S combining all the IOPs, calculated from Eq. (9), are also given.

| Layers | Depths [cm] | High Arctic                 |        |        |                             |      |                                   |        |        |        |  |
|--------|-------------|-----------------------------|--------|--------|-----------------------------|------|-----------------------------------|--------|--------|--------|--|
|        |             | <i>a</i> [m <sup>-1</sup> ] |        |        | <i>b</i> [m <sup>-1</sup> ] | g    | <i>b (1-g)</i> [m <sup>-1</sup> ] | S      |        |        |  |
|        |             | 480 nm                      | 540 nm | 600 nm | -                           |      |                                   | 480 nm | 540 nm | 600 nm |  |
| Snow   | 0-2         | 0.043                       | 0.0683 | 0.12   | 4277.1                      | 0.85 | 641.57                            | 0.008  | 0.010  | 0.014  |  |
| DL     | 2 - 20      | 0.043                       | 0.0683 | 0.12   | 410.44                      | 0.99 | 4.10                              | 0.102  | 0.128  | 0.169  |  |
| OII    | 20 - 57     | 0.043                       | 0.0683 | 0.12   | 396.73                      | 0.99 | 3.96                              | 0.104  | 0.130  | 0.171  |  |
| OII    | 57 – 97     | 0.043                       | 0.0683 | 0.12   | 291.53                      | 0.99 | 2.92                              | 0.121  | 0.151  | 0.199  |  |
| YII    | 97 - 137    | 0.043                       | 0.0683 | 0.12   | 97.90                       | 0.99 | 0.98                              | 0.205  | 0.255  | 0.330  |  |
| YII    | 137 - 177   | 0.043                       | 0.0683 | 0.12   | 48.40                       | 0.99 | 0.48                              | 0.286  | 0.352  | 0.446  |  |
| SL     | 177 – 185   | 0.043                       | 0.0683 | 0.12   | 276.52                      | 0.99 | 2.77                              | 0.124  | 0.156  | 0.204  |  |
| Seawa  | 185 –       | 0.0475                      | 0.050  | 0.12   | 0.89                        | 0.90 | 0.09                              | 0.571  | 0.660  | 0.758  |  |

ter

320

Figure 3: Irradiance measurements at the High Arctic site: vertical profiles of (a) downward planar irradiance, (b) upward planar irradiance, and (c) scalar irradiance. The second row shows the average cosines for respectively the downward, the upward, and the complete radiance values angularly defined (from left to right). In the last row, we see the net irradiance as a function of the depth in sea ice (g), the diffuse attenuation coefficient for the downwelling irradiance,  $K_d$  (in m<sup>-1</sup>), (h), and the Gershun's law derived absorption coefficient (i). For each subfigure, the three spectral band curves are displayed according to their colour. The broken lines are the measurement results, while the solid ones are the RT simulation outputs.

The first row of Fig. 4 shows azimuthally averaged radiance measured and modeled at multiple depths inside sea ice. Distributions at the surface are not shown as they were too much affected by hole effects and operator shadow. The angular radiance distributions are also not displayed at 180 cm and 200 cm depths because of uncertain camera vertical positioning.

The left to right columns (Fig. 4a, b, c) correspond respectively to the 480, 540 and 600 nm spectral bands. For comparisons of simulated and measured angular radiance distribution, we calculate statistical metrics such as the mean unbiased percent difference (MUPD [%]):

$$MUPD_{i} = 200 \cdot \frac{1}{K} \frac{1}{N} \cdot \sum_{k=1}^{K} \sum_{n=1}^{N} \frac{\left[ \overline{L}_{i,HL}(z_{k}, \theta_{n}) - \overline{L}_{i,Field}(z_{k}, \theta_{n}) \right]}{\left[ \overline{L}_{i,HL}(z_{k}, \theta_{n}) + \overline{L}_{i,Field}(z_{k}, \theta_{n}) \right]}$$
(11)

and the root-mean-square error (*RMSE* [%]):

$$RMSE_{i} = 100 \cdot \sqrt{\frac{1}{K} \frac{1}{N} \cdot \sum_{k=1}^{K} \sum_{n=1}^{N} \left[ \frac{\overline{L}_{i,HL}(z_{k}, \theta_{n}) - \overline{L}_{i,Field}(z_{k}, \theta_{n})}{\overline{L}_{i,HL}(z_{k}, \theta_{n})} \right]^{2}}$$
(12)

with  $\bar{L}_{i,HL}(z_k,\theta_n)$  being the HydroLight azimuthally averaged radiance while  $\bar{L}_{i,Field}(z_k,\theta_n)$  represents the field measurements at each discrete depth  $z_k$  [cm] and  $\theta_n$  the zenith angles between 0° and 180°. The average errors, given from Eq. (11) for the MUPD and Eq. (12) for the RMSE, are respectively 3.31 % and 10.33 % (blue channel), -3.90 % and 9.11 % (green channel), and -0.80 % and 10.08 % (red channel). At depths  $\leq$  60 cm, there are important differences between the measured and simulated angular radiance distributions near both angular extremities (20° and 160°). As we progress deeper inside the ice slab, these extremities errors seem to reduce as does the error curves at all zenith angles compared to their lower depth counterpart.

**Figure 4:** Depth resolved spectral angular radiance distributions azimuthally averaged. The first row represents the High Arctic radiance for (a) 480, (b) 540, and (c) 600 nm spectral channels, while the second row shows the same bands (in order from left to right) but for the Chaleur Bay site. The broken lines are the camera measurements and the full lines are the radiance data modeled with HydroLight (both at 1° angular resolution).

# 3.1.2 Chaleur Bay



One of the vertical profiles of spectral radiance captured at Chaleur Bay is displayed in the second row of Fig. 4. We show only the light field below the freeboard as the measurements above were particularly contaminated by the hole's effect on the light field. Radiances obtained with HydroLight RT simulations (see section 2.4 for the procedure) are presented in Fig. 4d, e, and f as plain lines for each depth of measurement. Table 2 shows the inherent optical properties inferred for the ice geometry. We inverted a 5 cm thick scattering surface layer with a b' of 72.85 m<sup>-1</sup> which decreases significantly to 5.55 m<sup>-1</sup> in the region

above freeboard. Next, two successive layers of interior ice (18-35 cm and 35-65 cm) with similar b' values of 5.59 m<sup>-1</sup> and 7.79 m<sup>-1</sup>, respectively. These values, for interior ice, are larger than what was observed in High Arctic. The last layer (before the water column) has a reduced scattering coefficient of 0.021 m<sup>-1</sup>. Apart from large scattering coefficients for interior ice, we also inferred high absorption coefficients from Gershun's law at that site. Above 35 cm, the coefficients are all higher than 1.80 m<sup>-1</sup>. Spectrally, the absorption is larger at 600 nm compared to 540 nm for almost all the layers (except for the skeletal layer), while the blue band coefficients surpass those in the red band in every layer. The high absorption of the medium is also reflected in the  $K_d$  coefficients which are higher than those of Arctic sea ice. The diffuse planar downwelling attenuation coefficients are shown in the supplemental document along with all the same quantities as the ones presented in Fig. 4 (see Fig. S1 and Fig. S2). The agreement between the RT simulated and measured zenithal radiances, quantified using the MUPD, are of -22.49, -11.26, and -15.62 % respectively for the 480, 540, and 600 nm channels. For the root-mean square errors, which gives a better idea of the residuals, we obtained 115.51, 91.48, and 106.86 % for the same spectral bands. These discrepancies larger than for the High Arctic inversion may be caused by more impact of the hole and self-shading on the measurements. As seen in the second row of Fig. 4, greater errors appear at low and large zenithal angles; regions in the radiance data prone to the impacts of the drilling hole and the shading of the stick we inserted in it. In addition, uncertainties associated with the ice thickness measurements taken during the fieldwork, as well as those related to the ice-ocean interface position in the model, may account for the notable differences at 70 cm. This is because the interface between the highly scattering sea ice and the ocean causes a large change in the shape of the radiance. Inaccurate positioning of the interface in the simulation will therefore lead to large differences in simulated and measured radiances. It should also be mentioned that the inherent optical properties inversion algorithm is based on global error minimization, i.e. it achieves a compromise where a greater error at a given depth may give rise to a smaller one in another region.




Table 2. Ice inherent optical properties obtained by fitting the HydroLight radiative transfer simulations to the radiance measurements at Chaleur Bay site (on 23 March 2022). The table provides the absorption coefficients a [m<sup>-1</sup>], the scattering coefficients b [m<sup>-1</sup>], the anisotropy coefficient of the phase function g, the reduced scattering coefficient b(1-g) as well as the similarity parameter S for all the layers. Below the surface slab, sits the drained layer (DL) then the interior ice (II) and the skeletal layer (SL).

|         | Depths       | Chaleur Bay                 |        |        |                             |      |            |        |        |        |  |
|---------|--------------|-----------------------------|--------|--------|-----------------------------|------|------------|--------|--------|--------|--|
|         | [cm]         |                             |        |        |                             |      |            |        |        |        |  |
| Layers  | _            | <i>a</i> [m <sup>-1</sup> ] |        |        | <i>b</i> [m <sup>-1</sup> ] | g    | b (1-g)    | S      |        |        |  |
|         |              |                             |        |        |                             |      | $[m^{-1}]$ |        |        |        |  |
|         | <del>-</del> | 480 nm                      | 540 nm | 600 nm | _                           |      |            | 480 nm | 540 nm | 600 nm |  |
| Surface | 0-5          | 2.20                        | 1.97   | 2.12   | 484.36                      | 0.85 | 72.85      | 0.171  | 0.162  | 0.169  |  |
| DL      | 5 – 18       | 2.20                        | 1.97   | 2.12   | 555.05                      | 0.99 | 5.55       | 0.533  | 0.512  | 0.527  |  |
| II      | 18 - 35      | 2.11                        | 1.80   | 2.02   | 558.86                      | 0.99 | 5.59       | 0.523  | 0.494  | 0.515  |  |

| II     | 35 - 65 | 0.87 | 0.65 | 0.77 | 779.14 | 0.99 | 7.79  | 0.317 | 0.277 | 0.300 |
|--------|---------|------|------|------|--------|------|-------|-------|-------|-------|
| SL     | 65 - 72 | 0.44 | 0.34 | 0.32 | 2.06   | 0.99 | 0.021 | 0.977 | 0.971 | 0.969 |
| Seawat | 72 –    | 0.22 | 0.11 | 0.18 | 0.87   | 0.90 | 0.09  | 0.847 | 0.748 | 0.821 |
| er     |         |      |      |      |        |      |       |       |       |       |

#### 3.2 Inversion errors




To evaluate any systematic error induced by the inversion procedure exposed in section 2.4, we generated a set of fifty artificial in-ice irradiance profiles from realistic depth dependent IOPs. The algorithm was then used to invert the optical properties or the different irradiance profiles using the following iteration numbers: N = 5, M = 8, and G = 150 (see section 2.4). These latter were determined empirically to reduce the error while limiting the computation time required to perform an inversion. The error for each layer is shown in Fig. 5, comparing the reduced diffusion coefficients found with the references. The mean absolute error for the inversion of the surface scattering layer, drained layer, interior layer, and sea water coefficients are 3.55, 12.23, 2.62, and 3.78% respectively. During the inversions, we observed that the algorithm converged towards the desired values, even if the starting values were far from the references.

**Figure 5:** Estimation of the errors made by the inherent optical properties (IOPs) inversion algorithm. The figure links the reference reduced scattering coefficients with ones estimated by the algorithm for the three layers ice model composed of a surface scattering layer, a drained layer, and an interior ice layer.

#### 4 Discussion







#### 4.1 Field measurements

Although we set several layers for each site (see section 2.4), it is interesting to note that subsequent layers merged into larger regions of constant scattering properties, suggesting zones of similar microstructure. The first two centimetres of pack ice are a special case, as they are made up of snow, the most scattering element, with the highest coefficient,  $b' = 641.47 \text{ m}^{-1}$ . Just below, sit the drained layer (DL) constrained by the freeboard position. This layer is also known to cause significant scattering as the inclusions are drained and the ice more porous. The value of the reduced scattering coefficient for this layer is 4.10 m<sup>-1</sup> 1, which fall within the previously reported range of 2.14-12.0 m<sup>-1</sup> (Light et al., 2008). The two subsequent slabs (20-57, 57-97 cm) below freeboard have also considerable inverted b' of 3.96 m<sup>-1</sup> and 2.91 m<sup>-1</sup>. These values are larger than those inferred for multiyear interior ice during summer of 1998 in the Beaufort Sea, which range between 0.5-1.8 m<sup>-1</sup> (Light et al., 2008). They however fall in the gaps of larger b' of 2.1-4.4 m<sup>-1</sup> (between 6-76 cm) and 2.8-7.1 m<sup>-1</sup> (between 10-100 cm) measured respectively for snow covered and bare first-year interior ice (Perron et al., 2021). Below 97 cm, we observed b' inside the Light et al. (2008) interior ice (II) interval as our values are within 0.48-0.98 m<sup>-1</sup>. In the last ice layer (177-185 cm), we notice an increase of the scattering coefficient. This is probably the skeletal layer, formed by the advective exchange between ocean and ice. This leads us to assume that the ice at that High Arctic site was probably composed of two types of more translucent interior ice; an old interior ice (OII) from 20-97 cm and a younger interior ice (YII) from 97-185 cm. MY ice is known to have large variations in its optical properties and in boundaries of its different layers due to multiple melt and growth cycles (Pegau and Zaneveld, 2000). This likely means that the YII was recently formed, while the OII layer was shaped during previous summers, giving time for brine pockets to be drained over one or multiple melt seasons (Perovich et al., 2002). It would indeed have been very interesting to explain the number of seasons the ice had survived. However, this would have required crystallographic investigations accompanied by oxygen isotope analyses and could be the subject of a future study. Higher scattering in the upper half of the ice column was also evident in optical observations and investigations conducted a few hundred meters away on the same ice floe (Katlein et al., 2021). From this last study, a four-layer model using DORT 2002 RT model (Edström, 2005) helped inverting reduced scattering coefficients of 25 m<sup>-1</sup> for the surface scattering layer and 2.5 m<sup>-1</sup> for the interior ice. This latter b' falls near the center value of 2.22 m<sup>-1</sup> of our inversion for interior ice (0.48 m<sup>-1</sup> 3.96 m<sup>-1</sup>).

For the absorption coefficients of High Arctic MY sea ice derived through Gershun's law (Fig. 3i), the depth median absorption coefficient of 0.061, 0.054, and 0.138 m<sup>-1</sup> are close to those for pure bubble-free ice of 0.043, 0.0683, and 0.12 m<sup>-1</sup> for the 480, 540, and 600 nm channels respectively. This would suggest that using the pure ice assumption would be valid, particularly in this case, as it is reasonable to assume that sediment is unlikely to be found in multi-annual sea ice cores near the North Pole. The larger median coefficient in the blue compared to the green spectral band may be surprising but agrees with previously reported values found in landfast sea ice (Ehn et al., 2008b). In the upper region ranging to 80 cm, we notice larger a in the green band compared to the 480 nm channel, while for depths  $\geq$  100 cm, the blue absorption coefficients surpass those in the

green (see Fig. 3i). This is apparent also in the color of the ice of Fig. 6b – extracted from jpeg images saved along the raw ones – which displays blue colors becoming greener at 100 cm. The increase of a for shorter wavelength could be caused by algal or non-algal particles (AP and NAP) that may be larger in concentrations as we progress toward the ice-ocean interface (Ehn et al., 2008b). The larger surface values are hard to explain physically as the presence of air in the snow should slightly reduce the absorption coefficient, a. We think that this may be due to hole effects, a bright one at the surface, increasing 445 downwelling irradiance and a dark spot down under decreasing the upwelling irradiance. Regarding the negatives inverted absorption coefficients caused by the increase in net irradiance (see Fig. 3g), we believe that measurements artefacts may be responsible. They include possible angular misalignment of the optics inside the hole or surface effects such as shadow of the operator near the hole or snow displacement. The average cosine  $\bar{\mu}$  has the same spectral trends as the measured absorption coefficients, as they are related through Gershun's law, presented in Eq. (5. The increase of  $\bar{\mu}$  with depth starting from 60 cm 450 (see Fig. 3f) reflects higher proportion of light rays that vanished at large angles the deeper we are in ice. This is explained by being closer to the ice-ocean interface where there are less photons going upward due to the significantly less scattering in the last ice layers and in the ocean. This is also why the gradient with depth for  $\bar{\mu}_u$  is larger than  $\bar{\mu}_d$ , as the light rays in the downwelling field are less affected by the proximity of the ice-ocean interface. No in-ice average cosine measurements were found in the literature, but some reported  $\bar{\mu}_d$  just below sea ice bottom ranging from 0.59 to 0.70 based on direct observations 455 (Katlein et al., 2014; Massicotte et al., 2018; Matthes et al., 2019) or models (Arrigo et al., 1991; Ehn and Mundy, 2013). Downwelling average cosines measured by Matthes et al. (2019) show spectral  $\bar{\mu}_d \leq 0.59$ , 3.0 m below bottom for an ice covered with snow. These measurements reported under ice are consistent with the one we report here at 15 centimetres under ice 0.556, 0.555 and 0.562 for the 480, 540 and 600 nm bands respectively. Also, measurements of the mean cosine in the part of the ice where several organisms live could have an impact on calculations of primary production in the ice. Primary production models could gain in accuracy if they included the value of  $\bar{\mu}$  in their calculations. Algae, like the radiometric tool used in this study, are sensitive to the entire radiative field, whether it comes from above or below.

At the Chaleur Bay site (72 cm ice thickness), the first layer which includes the granular layer, has the largest reduced coefficients b' of 72.85 m<sup>-1</sup> which is significantly smaller what was observed in high Arctic for the same layer. Absence of snow at the site may explain this lower value. The two interior ice layers (18 - 35 cm, 35 - 65 cm) have b' of 5.59 m<sup>-1</sup> and 7.79 m<sup>-1</sup> respectively. Those layers scattered in the same range of the layer above freeboard ( $b' = 5.55 \text{ m}^{-1}$ ) and their values are slightly higher than past observations in interior ice (and those presented in this work in High Arctic). From our discussions with a local ice fisherman, we understood that heavy and multiple snowfalls during winter had come to melt and refreeze, forming layers of coarse ice grains. These are superimposed ice layers (Ehn et al., 2008b) and would explain the larger scattering coefficients inverted in the zones below the freeboard. Non-constant absorption coefficients with depth allowed good agreements between the simulations and the radiometric observations. The ice color for each depth displayed in Fig. 7c also supports variation of the coefficients with depth as the RGB vary greatly depending on position within the slab. The inferred a ranges from 0.32 to 2.20 m<sup>-1</sup> across the entire ice column. Above water level (18 cm), the absorption coefficients



stay constant in both simulated layers (0-5 cm, 5-18 cm) at  $2.20 \text{ m}^{-1}$ ,  $1.97 \text{ m}^{-1}$ , and  $2.12 \text{ m}^{-1}$  for respectively the blue, green, and red bands. Although being quite large, we think that these values are plausible as some studies reported absorption coefficients over  $1.0 \text{ m}^{-1}$  in the visible spectrum inside sea ice of Liaodong Bay; region surrounded by some industrial, agricultural, and residential zones (Xu et al., 2012b). In the first layer below freeboard, the absorption coefficients decrease slightly and then more rapidly in the two last layers up to a in the range of  $0.32 - 0.44 \text{ m}^{-1}$ . These latter inverted coefficients are higher than what was reported for pure interior ice. This informs us that the ice at Chaleur Bay may have contained higher concentration of algal and non-algal particles than the Arctic site. The higher absorption coefficients in the blue band compared to the green channel (see Table 2) are consistent with Fig. 6c, which shows greener colors compared to those observed for the High Arctic (Fig. 6b).




**Figure 6:** (a) Summary of the similarity parameter *S* inverted from measurements at both High Arctic (solid lines) and Chaleur Bay sites (dashed lines), and those reported in previous studies for different types of sea ice. The background rectangles represent past measurements [32,33,69] while the curves are the depth dependent similarity parameters in each spectral band and for both sites. To calculate the similarity parameters of Light et al. (2008) and Perron et al. (2021), we used the absorption coefficient of pure ice at 540 nm [65]. The *S* of Ehn et al. (2008) are provided in their paper at 500 nm. (b)-(c) RGB color of the High Arctic and Chaleur Bay ice respectively extracted from jpeg images saved along DNG files (see Fig. S4 in the supplemental document for example of these rectilinear images). Obtaining these two 1D images involved: 1) taking the average pixel value inside a square at the center of the image, 2) normalization by the diffuse planar downwelling attenuation coefficient spectrally averaged, 3) normalization by the maximum.

For highly scattering media such as the ice of the studied sites, the inverted IOPs are only true under the similarity principle as two different sets of IOPs with the same S give similar light fields. Following Eq. (9), if the ratio b'/a (or b/a given constant asymmetry parameter) increases, the similarity parameter decreases. In other words, the higher the value of S, the greater the energy loss in the layer concerned due to absorption. Figure 6a shows the similarity parameter profiles for the two

field campaigns at the three measured wavelengths. The background shows the similarity parameters (540 nm) as calculated using measurements found in the literature. Firstly, this visualization effectively summarizes the variation within the profile of the optical properties of High Arctic ice. Indeed, low values of S can be seen at the top of the profile, reflecting the dominance of scattering in this region. Throughout the profile, the value of the similarity parameter increases. Indeed, scattering decreases and absorption increases in the lower layers. Spectrally, we can see that the value of S is greater for red than for blue, which is to be expected as ice absorbs the blue part of the visible spectrum to a minimum. In short, the profile of the similarity parameter for measurements in the High Arctic shows that the ice there appears to be rather pure, and that scattering loses its importance significantly as we move down the profile. In the Chaleur Bay data analysis, the case is less that of pure textbook sea ice. In this case, there are two distinct zones: the upper zone, where absorption dominates, and the lower zone, where the value of S decreases. This behavior can be explained by the phenomenon of ice formation at this point. Indeed, discussions with residents have confirmed that pack ice in this area forms in two stages. First, during the first sustained subzero temperatures, a layer of ice about 30 cm thick forms. This would appear to be the layer with the lowest S values. As the winter season progresses, heavy snow accumulations weigh on the pack ice, leading to the formation of superimposed ice. Presumably, when this snow is flooded and frozen, it has already accumulated atmospheric deposits of dust. This would explain the second part where the S values are lower. Spectrally, S values are arranged differently for the High Arctic. In fact, the S value for blue is greater than for green and red, reflecting greater absorption in the shorter wavelengths of the visible spectrum. One possible explanation for this spectral behavior could be the presence of chlorophyll, which absorbs most of the blue part of the spectrum, as well as a little of the red, leaving out the green. This would explain why the value of the similarity parameter is highest for blue, then red and finally green. Figure 6b shows that, as described above, the ice has a bluish hue, as would be expected for ice from the High Arctic. Figure 6c, on the other hand, shows a greyish hue that might be expected from dirty snow, followed by an increasingly greenish tinge, as would be expected from the presence of large numbers of photosynthetic organisms. In the future, analysis of particulate absorption and chlorophyll measurements could give rise to the possibility of confirming the insight we obtain from the mean color of the ice.

# 520 **4.2** Errors analysis






Certain sources of uncertainties may have affected the IOPs inversion from the set of measurements presented in this study. These included uncertainties due to the measuring device and its manipulation on-field, source of errors due to the 3-D variability in ice structure itself, and HydroLight inversion uncertainties. First, the absolute calibration of the radiance camera bears uncertainties, discussed in details in Larouche et al., 2024b. Second, potential manipulation errors that could have occurred during fieldwork are the following: misalignment of the optics inside of the hole leading to misaligned angular coordinates of the radiance values, shadow of the manipulators affecting surface measurements, and unprecise depth determination as sometimes the holding stick came up folded. Another instrument artifact affecting the geometric light field is the presence of the camera-stick assembly inside the ice (Picard et al., 2016). We did a brief investigation of self-shading effects that reveals that radiance distributions are affected more strongly at small and large zenithal angles. This likely explains

why the simulated and measured angular radiance distributions are more distant at the two extremities of the curves as seen in Fig. 4. In addition, the solid nature of the ice leaves no choice but to dig a hole for internal measurements. From above or below the camera position, the hole creates path for light to easily propagate and add to the fluxes that there would be without the opening. These hole effects increase when the camera is positioned near the boundaries, as the hole is seen inside a larger portion of the FOV. A second source of errors due to ice itself is the large spatial variation in its structure (micro and macro scale) over only a few meters (Frey et al., 2011; Pegau and Zaneveld, 2000; Perovich and Gow, 1996). The light field from a nearby region can affect the radiance at the position of interest. This is difficult to discern from our radiance measurements and cannot be simulated in HydroLight. To be able to characterize horizontal variability effect, several measurements with the camera could be taken. For example, it would be interesting to perform a transect with the camera, gradually approaching a melt pond, and observe the distance at which the light field begins to deviate from azimuthal homogeneity. Finally, there is also the anisotropic scattering coefficient that was neglected in this study – as poorly known and not configurable in the HydroLight RT model – and which might have permitted better IOPs inversion and fitting of the field observations.







The scattering coefficient inversion algorithm performed well for the evaluation dataset. The error for the layers is around 3%, except for the drained layer where the average error of the algorithm is four times larger at 12%. This error is probably due to the sensitivity of the loss function to scattering events occurring in this layer. Indeed, although scattering less, this layer is smaller (27.5 cm) than the inner layer (117.5 cm) and is ten times less scattering than the surface scattering layer. It is reasonable to assume that the weight of the inner and surface layers dominates the loss function because they have greater optical depth, which explains the lower performance for the drained layer. Nonetheless, the algorithm we developed was able to find the values we were looking for, even if they were very far from the starting values. This performance is due to the algorithm ability to explore the solution space and find the global minimum. This exploration of the space is made possible by the iterative process, where the search is repeated several times. The use of reflectance as a loss function in that first step also helps to explain the algorithm good performance. In fact, the reflectance of a layer is little influenced by other surrounding layers, making this measure more robust. Unlike reflectance, irradiance measured at a given location is highly dependent on the layers above and below it. This strong inter-layer dependence makes inversion very difficult, as the loss for the different layers is interdependent. To sum up, using the recursive method with reflectance provides a good global exploration and enables us to find the vicinity of the global minimum. The second more sensitive step enables fine convergence when the first step provides a profile very close to the desired solution.

On the relevance of angular radiance distribution measurements, it has been determined that we could successfully invert the IOPs having simultaneous measurements of  $E_d$ ,  $E_u$ , and  $E_o$ . Ultimately, by making the observed and simulated irradiances match, we noticed correspondences when verifying radiances distributions. This is typical of the diffusion regime where the zenithal radiances approach an asymptotic shape decaying with depth at the same rate as the irradiances (Preisendorfer, 1958). The usefulness of radiance over almost  $4\pi$  steradians resides in that we can collect all the possible irradiance quantities ( $E_d$ ,  $E_u$ ,  $E_d^o$ ,  $E_u^o$ , and  $E_o$ ) simultaneously from one capture of the 360-degree camera. To gather all these quantities from currently available radiometers at the same time would be a complicated task as they are usually designed to measure only one type of irradiance. Packaging this into a single profiler would require large assemblies of tens of centimeters. In addition, as we would capture radiance closer to boundaries with non-isotropic incident light field, in layers with small scattering coefficients, the azimuthal and zenithal angular shape of the light field would be of great interest. These conditions could happen in the marine environment, for instance inside blue ice of glacier crevasses and icebergs (Warren et al., 2019). For future studies, it would also be interesting to investigate in more detail the range of scattering coefficients for which the shape of the phase function has an impact on angular radiance distributions. This would lead to a better understanding of the need to increase the number of phase function moments in radiative transfer simulations in order to correctly model the observations. In these cases, the angular shape of the radiance would contain information on the phase function.

#### 5 Summary








In this paper, we successfully demonstrate the utility of 360-degree cameras to study in-ice radiative transfer. Those compact optical systems enable capture of full radiance angular distributions at fixed point in space and inside three spectral bands centered on 480, 540, and 600 nm. We would like to emphasize on this new possibility to recover of all the irradiance radiometric quantities  $(E_d, E_u, E_d^o, E_u^o, \text{ and } E_o)$  from unique captures of this low-cost camera. Subsequently, from these irradiances, we were able to retrieve profiles of IOPs using an inversion algorithm that matched HydroLight simulated radiometric quantities to those measured at two sites: in High Arctic and Chaleur Bay (Quebec). In the 1.85 m thick High Arctic MY ice, we inferred reduced scattering coefficient of 641.57 m<sup>-1</sup> at the surface, and three distinct regions of interior ice: older interior ice (OII) with b' between  $2.92 - 3.96 \,\mathrm{m}^{-1}$ , younger interior ice (YII) with b' within  $0.48 - 0.98 \,\mathrm{m}^{-1}$ , and a skeletal layer with a reduced scattering coefficient of 2.77 m<sup>-1</sup>. Inside the seasonal Chaleur Bay interior ice, significantly higher light attenuation was assessed, due to both larger absorption,  $0.32 - 2.11 \text{ m}^{-1}$ , and more scattering,  $0.021 - 7.79 \text{ m}^{-1}$ , compared to the High Arctic site. Those results may be attributed to the presence of superimposed ice as well as absorbing particles (AP and NAP). The inversion problem faced in this work was greatly underdetermined. We had very little complementary information about the ice that forced assumptions for the absorption coefficients and the asymmetry parameters g. Combined with instrumental and manipulation errors on the field, the optimization algorithm – although being quite accurate for perfectly generated angular radiance distributions with relative errors between 2.62 and 12.23 % – may have converge toward erroneous scattering coefficients. This is why we prefer reporting the similarity parameter S which is right for the observed radiometric quantities. Much more effort continues to be required in order to improve the inversion. For future fieldwork, we intend to use independent measurements of absorption coefficients (total, algal and non-algal particulate matters) from melted co-localized sea ice layers as well as in situ measurements of b' with a diffuse reflectance probe (Perron et al., 2021). These measures could enable significant improvement of the inversion process. Measurements of ice core temperature, salinity, and birefringence image of the crystals could also be captured. This would give us a more complete story of the sea ice microstructure and allow better assessments of separated *a*, *b*, *g*. Finally, two engineering challenges remain. Since each camera is currently calibrated individually, it would be valuable to purchase a batch (e.g., more than 10) of identical cameras to assess the variability introduced during manufacturing. If this variability proves to be low, it could reduce or even eliminate the need for systematic calibration, making the camera easier to use and enabling widespread adoption within the research community. Additionally, the camera must be removed from the hole between each measurement to allow time for image acquisition. Under normal conditions, it could be controlled remotely via radiofrequency signal (Wi-Fi or Bluetooth) sent from a smartphone. However, the ice pack completely absorbs these frequencies. It would therefore be valuable to develop a solution, that remains as compact as possible, to control the camera from the surface. Such a system would accelerate data acquisition and minimize the risk of disturbing the environment.




This study has demonstrated the potential offered by the new 360-degree cameras available on the market. These cameras can replace the research prototypes very cumbersome to develop, and potentially presenting lower performance. The commercial camera used here was, among those available on the market, one of the most suitable for our application. The smartphone industry continuing to strongly stimulate innovation in optical imaging, new camera models will undoubtedly become available in the future, smaller, more sensitive, and perhaps more effective in their spectral resolution. Our study has opened the door to the use of these highly advanced commercial technologies for characterizing the optical properties of sea ice.

# Appendix A

Legendre polynomials form a set of infinite orthogonal functions with interesting mathematical properties. They were previously used to fit angular radiance distributions (Kattawar, 1975). These polynomials are also employed for discretization of radiance in the numerical discrete-ordinate solution for the radiative transfer equation (Stamnes et al., 1988). They were naturally selected for the extrapolation of the unknown radiance values in water (camera below ice freeboard). The spectral radiance  $\bar{L}$  [W sr<sup>-1</sup> m<sup>-2</sup> nm<sup>-1</sup>] at a given depth and wavelength is expressed as

$$\bar{L}(\mu) = c_0 + \sum_{l=1}^{l=5} c_l \cdot P_l(\mu)$$

$$P_l(\mu) = \begin{cases}
\mu, & l = 1 \\
\frac{1}{2}(3\mu^2 - 1), & l = 2 \\
\frac{1}{2}(5\mu^3 - 3\mu), & l = 3 \\
\frac{1}{8}(35\mu^4 - 30\mu^2 + 3), & l = 4 \\
\frac{1}{8}(63\mu^5 - 70\mu^3 + 15\mu), & l = 5
\end{cases}$$
(A1)

with  $\mu=\cos\theta$ ,  $P_l$  and  $c_l$  being respectively the Legendre polynomials and their corresponding coefficient of degree l. A least-square minimization routine led to the determination of the Legendre coefficients for the azimuthally averaged spectral radiance. This is possible knowing that azimuthal dependence of the light field under optically thick ice is negligeable (Pegau and Zaneveld, 2000). Angles below 25° ( $\mu>0.906$ ) were discarded to mitigate the effect of self-shadow and the drastic light fall-off at the edge of the FOV. We used the development of the polynomials up to a fifth degree l, found sufficient to fit different shapes of  $\bar{L}(\mu)$  commonly found in the medium. Figure S3 of the supplemental document shows some results of the fit for the data taken during AO2018 expedition. Two example cases are displayed: for a depth of 40 cm at  $\lambda=480$  nm and for a 120 cm at  $\lambda=540$  nm.

#### Appendix B

This section is aimed at potential users interested in making radiometric measurements with a commercial 360-degree camera in scattering environments. We focus on the challenges raised by changes in incident radiation and camera disturbances to the radiative field within ice.

Firstly, it is important to minimize disturbance to the incident radiation by avoiding movement between the sun and the site of interest and by reducing sources of shading. Ideally, the site should be as homogeneous as possible over a large area, particularly in terms of snow cover. It is also essential to use an upward-looking radiometer to monitor irradiance throughout the experiment, especially under variable sky conditions such as passing clouds. Secondly, the radiative field within the pack ice must remain as undisturbed as possible, despite the hole made for the measurement. The use of a diffuser to block unwanted light should be considered. Additionally, the camera and boom should be painted to match the reflectance of the surrounding medium, to avoid absorbing light that would alter the radiative field. Thirdly, measuring light in air above the freeboard level presents several challenges due to the reduced radiance caused by the change in refractive index. A satisfactory solution to this problem has yet to be identified.

# 640 Code availability

The scripts used in this study are available in the online repository:

https://github.com/RaphaelLarouche/radiance\_camera\_insta360. The latest version is also archived in the Zenodo repository 10.5281/zenodo.4660993 (Larouche, 2024). This repository includes all scripts for processing fieldwork data and routines for generating the figures presented in this paper. Note that the codes are subject to continuous development.

# 645 Data availability

The dataset of this study is accessible online through the Zenodo repository: <a href="https://doi.org/10.5281/zenodo.14263255">https://doi.org/10.5281/zenodo.14263255</a> (Larouche et al., 2024a).

Additionally, data from RAMSES irradiance sensors collected during the AO2018 expedition are accessible via the Meereisportal platform: <a href="https://www.meereisportal.de/en/">https://www.meereisportal.de/en/</a>. These datasets can be retrieved from <a href="this page">this page</a> under the buoy name 2018R4 (Grosfeld et al., 2016).

# **Executable research compendium (ERC)**

#### **Author contribution**

C.K. and B.R. did the fieldworks and acquired the in-ice measurements presented in this study. R.L., S.L.G., and C.K. designed the camera calibration and characterization methodologies, while R.L. carried them out. B.R., R.L, C.K. and S.L.G. designed and performed the radiative transfer simulations. R.L and B.R. prepared – with equal contribution – the first draft of this manuscript, with inputs from all the authors. S.L.G. and M.B. had the idea to use 360-degree camera inside sea ice and provide help and guidance throughout the project. S.T. and M.B. supervised the project and provided guidance.

#### **Competing interests**

The authors declare that they have no conflict of interest.

#### Acknowledgements

Raphaël Larouche was supported by the SMAART program through the Collaborative Research and Training Experience program (CREATE) of the Natural Sciences and Engineering Council of Canada (NSERC). This research was supported by the Sentinel North program of Université Laval, made possible, in part, thanks to the funding from the Canada First Research Excellence Fund, the Canadian Excellence Research Chair on Remote sensing of Canada's new Arctic frontier, and Marcel Babin discovery grant #RGPIN-2020-06384. We also gratefully acknowledge the scientific and financial support of Québec-Océan. Field measurements during the AO18 expedition onboard icebreaker Oden were funded by the Helmholtz infrastructure initiative "FRAM" (Frontiers in Arctic marine Monitoring) and supported by the Swedish Polar Research Secretariat (SPRS). We thank Philipp Anhaus and Mario Hoppmann for assistance in the field. CK was supported by a postdoctoral fellowship from Sentinel North. We would like to thank Anne-Sophie Poulin-Girard for useful inputs regarding calibration methodologies, Marie-Hélène Forget for coordination, Numerical Optics CEO Dr. John Hedley for his generous answers to our countless

questions about HydroLight. Also, we would like to thank Prof. Daniel C. Côté, Yasmine Alikacem, and Christophe Perron for fruitful discussions.

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

# Supplemental document: Inferring Inherent Optical Properties of Sea Ice Using 360-Degree Camera Radiance Measurements

**Figure S1:** Vertical profiles of all irradiance quantities (planar downward, planar upward, and scalar) measured and simulated in the 480 nm (a), the 540 nm (b), and the 600 nm (c) spectral channels. In the same order of wavelengths, the second row shows the average cosines (planar downward, planar upward, and scalar) as a function of depths across the different boundaries measured during the fieldwork.

Figure S2: In order of rows from top to bottom, the figure displays the net planar irradiances, the diffuse attenuation coefficients for the downwelling irradiance,  $K_d$  (in m<sup>-1</sup>), and the Gershun's Law derived absorption coefficients. The observations are presented as circle markers while the HydroLight simulations are shown as plain lines. The rows from left to right are respectively the 480 nm, 540 nm, and 600 nm spectral bands.

Figure S3: Legendre polynomials fit results on the azimuthally averaged angular radiance distributions measured in High Arctic. Two radiance distributions example are shown: at a depth of 40 cm and at  $\lambda = 480$  nm, and at 120 cm for the band centered on 540 nm. The first row displays the Legendre polynomials curves fitted over the zenithal radiance (with the Legendre coefficients  $c_1, c_2, ..., c_5$ ) for (a) the first and (b) the second case. The second and third rows display (in order of case) the raw angular distributions for all the zenithal and azimuthal

directions (c)-(f), the Legendre polynomials curves re-projected over all the azimuth 360° (d)-(g), and the relative errors between the latter (e)-(h). The average and standard deviation of the relative errors |e| are respectively 6.2 % and 5.4 % for figure (e) and 8.6 % and 6.0 % for figure (h).

**Figure S4:** Fisheyes jpeg circular images (saved along DNG raw files) transposed into equirectangular grids. The images are shown columnwise for multiple depths within sea ice at (a) High Arctic (185 cm thick ice) and (b) Chaleur Bay sites (80 cm thick ice). These equirectangular grids have longitudes (FOV in x) x latitudes (FOV in y) less than 360° x 180° as acquired below the waterlines. We can interestingly observe greener colours in the Chaleur Bay ice (b) compared to bluer ones in Arctic sea ice (a).