# Peer review of "Inferring Inherent Optical Properties of Sea Ice Using 360-Degree Camera Radiance Measurements"

_EGUsphere, 2024_

## Author Comment (AC1)

**Response to reviewer # 1**

RC1: 'Comment on egusphere-2024-3819', Anonymous Referee #1, 28 Mar 2025 reply

We would like to thank reviewer #1 for taking the time to read our research article and to give us his insightful comments. We appreciate the constructive comments enabling us to improve our manuscript. Responses to the comments are given below.

**This manuscript presents a novel technique for investigating in situ magnitude and shape of the radiation field within a sea ice cover. Application of a commercially available 360 deg camera, along with sophisticated radiative transfer modeling, is demonstrated to yield detailed, vertically-resolved inherent optical properties of sea ice. The topic of this manuscript is of high interest to the TC readership and the manuscript is clear and concise, with appropriately illustrative figures. I found the manuscript a pleasure to read and think it is publishable in something very close to its present form. My only comments are minor, as detailed below. I found it particularly interesting to see the skeletal layer resolved in this study.**

**Fig 1c: Could be a bit more clear about the camera FOV and what it is seeing. The fact that the FOV is reduced to 76 deg (water, compared to air) should be mentioned in the text, not strictly relegated to a figure caption. It's not clear from the drawing in Fig. 1c where exactly the two fish-eye lenses view, nor is it clear what the solid angle 2*pi sr refers to.**

We agree that additional details on the camera's field of view and viewing direction should be explicitly included in the text. To address this, we have modified the sentence at lines 109-110 as follows:

"The Insta360 ONE low-cost omnidirectional camera (see Fig. 1c) has a diameter of 5 cm and includes two fixed-aperture ($f\# = 2.2$) fisheye lenses."

Additionally, we have incorporated the following clarification at lines 112-116:

"When the camera is in air, each imaging sensor can capture light from a hemispherical solid angle of $2\pi$ steradians. In water, this solid angle is reduced due to a decrease in the field of view along the optical axis of each lens, from 90˚ to 76˚ (Larouche et al., 2024b). In our measurement set-up, the optical axes (aligned with the $zc$ axis in the zoomed region of Fig. 1c) of both fisheye lenses are oriented 90˚ from the zenith."

**167: "In ice…", would be good to be more explicit, "In sea ice…"**

"In ice" was replaced by "In sea ice" as suggested.

**Figure 2: lots packed in here, would be helpful to have a bit more orientation (authors have been looking at these distributions, but they are new to us readers). E.g. line 274, help me see "is apparent" by walking me through the distribution in the figure.**

Commented [Ui1]: The azimuth angle is set relative to what?

Thank you for your comment. This part was added in the lines 273 – 280:

"These polar graphs show radiance angular distribution in spherical coordinates. The azimuth angle corresponds to a fixed reference on the camera, with lens #1 arbitrarily set at 90 degrees and lens #2 at 270 degrees when transforming RAW images into radiance. However, as the radiative field in sea ice is considered to be homogeneous as a function of azimuth angle, the camera is not positioned in exactly the same way for each measurement. This could have an impact in ice with very low scattering or under a melt pool, but not in the two cases studied. The zenith angle indicates where radiance comes from relative to the vertical axis. It varies from 0 (center of graph, downward direction) to 160 degrees (outer ring, upward direction) and indicates the elevation of the energy direction where 0 degrees indicates a downward direction (towards the ocean) and 180 degrees would indicate a perfectly upward direction (towards the atmosphere). The top panels (a, b and c) show radiance with the same color scale for the three channels centered at 480, 540 and 600 nm. The signal is predominantly blue in a downward direction (center of graph), followed by green and red. In the bottom panels (d, e and f), radiance is normalized to each depth, allowing us to better appreciate how its shape changes with depth. At higher elevations, the signal is much more homogeneous, whereas deeper within the sea ice, the angular distribution of radiance becomes increasingly downward. This effect is further accentuated at longer wavelengths."

**245: Eqn 9 exponent is – ½ (difficult to see negative sign, but so important!)**

The exponent has been rewritten to better highlight the negative sign.

**305: "imply fieldwork error" how about "likely derive from large observational uncertainties"?**

Thank you for your comment. We changed the sentence accordingly.

**347: is "zenithal" a word?**

Yes, "zenithal" is a recognized adjective, as defined in the Merriam-Webster dictionary. Given its recognition, we have opted to retain it in the manuscript.

**354: "Gershun's law"**

Thank you for your comment, the error has been corrected.

**395: "The first two centimetres of pack ice are a special case, as they are made up of snow…" Snow or surface scattering layer?**

The sampled multi-year ice was covered at that moment by 2 cm of snow. There could have been a surface scattering layer, but it was not observed in that specific case.

**409: winters? Or previous summer? Multiple melt seasons? Or one previous melt season?**

Thank you for your comment. In fact, we wanted to say that the ice had undergone several seasonal cycles, both expelling brine during the cold freeze-up period and draining during melt periods. It would indeed have been very interesting to explain the number of seasons the ice had survived. However, this would have required crystallographic investigations accompanied by oxygen isotope analyses. This could be the subject of a future study.

**556: spectral bands centered on 480, 540, 600 (since they likely aren't strictly at those wavelengths)**

Thank you for pointing out this inaccuracy, it has been corrected in the text.

**563 – 564: "significantly higher light attenuation was assessed, due to both larger absorption, 0.32 – 2.11 m-1, and reduced scattering coefficients, 0.021 – 7.79 m-1," . Here "reduced scattering coefficients" is confusing—it refers to b', but it also sounds like the b' values are lower, when I don't think that's the intent. Rewrite for improved clarity.**

Thank you for pointing this out. It is true that using the term "reduced" leads to ambiguity. The sentence has been modified to say simply, "more scattering".

---

## Author Comment (AC2)

**Response to reviewer # 2**

**RC2: 'Comment on egusphere-2024-3819', Anonymous Referee #2, 18 Apr 2025**

The manuscript presents a novel method for measuring the inherent optical properties (IOPs) of sea ice at varying depths, using RGB imagery to capture the light field. This technique is demonstrated on two distinct types of sea ice and shows potential for resolving vertical features such as surface scattering and skeletal layers. The approach is an innovative and compelling adaptation of commercial technology for cryospheric research. It is well-suited for publication in The Cryosphere.

Overall, the paper is clearly written and logically organized. Only minor revisions are needed prior to publication. While I have experience with sea ice AOPs and found the equations and methods clearly presented, I defer to other reviewers with expertise in IOPs and radiative transfer for a more rigorous assessment of the methodological details.

We would like to thank the reviewer for taking the time to read and provide valuable comments on our article. The comments initiated interesting discussions among the authors of this article and, we believe, helped improve the quality of the outcome. General comments are discussed below, along with minor comments and editing suggestions.

**General Comments:**

- **Clarification of Surface Layers (Chaleur Bay):**

  The manuscript could more clearly identify whether there is snow, snow-ice, or surface scattering layer (or some combination), particularly at the Chaleur Bay site. For example, Line 394–395 refers to the top ice layer as "snow," whereas elsewhere it is described as a surface scattering layer. Lines 449–452 also reference possible snow-ice. Greater clarity regarding the origin of the surface layer (or acknowledgment of uncertainty) is important for situating the findings in the context of other IOP studies.

Thank you for your comment. You're right that it's essential to be able to compare the measurements reported in this article with those published in the literature. As discussed below, we were unable to distinguish whether the surface layer at Chaleur Bay was sun-transformed ice, snow that had undergone several freeze-thaw cycles or a surface scattering layer. Unfortunately, this is often the case in spring ice measurements.

The aim of this article is to present an innovative instrument for measuring the apparent and inherent optical properties of two very different types of ice: one that we have been able to assume to be pure (High Arctic) and another containing many

impurities (Chaleur Bay). These two cases raise distinct methodological challenges, which is why we have decided to present them despite their imperfections. It's true that oxygen isotope measurements, crystallographic analysis or photos of carrots could have helped put the results into context.

- **Guidance for Future Applications:**

  **The manuscript nicely lays groundwork for future work by sharing code and methodology. However, it would benefit from a more specific summary of practical considerations for field deployment. For example, what engineering challenges remain? A brief section or appendix summarizing guidance for future users of the method would be valuable.**

Thank you for your comment. We agree that we need to share with the community all our learnings as well as the remaining deployment and engineering challenges. We added the following text at the end of the manuscript (lines 594-600).

"Finally, two engineering challenges remain. Since each camera is currently calibrated individually, it would be valuable to purchase a batch (e.g., more than 10) of identical cameras to assess the variability introduced during manufacturing. If this variability proves to be low, it could reduce or even eliminate the need for systematic calibration, making the camera easier to use and enabling widespread adoption within the research community. Additionally, the camera must be removed from the hole between each measurement to allow time for image acquisition. Under normal conditions, it could be controlled remotely via radiofrequency signal (Wi-Fi or Bluetooth) sent from a smartphone. However, the ice pack completely absorbs these frequencies. It would therefore be valuable to develop a solution, that remains as compact as possible, to control the camera from the surface. Such a system would accelerate data acquisition and minimize the risk of disturbing the environment."

And we added this text into appendix (see Appendix B):

"This section is aimed at potential users interested in making radiometric measurements with a commercial 360-degree camera in scattering environments. We focus on the challenges raised by changes in incident radiation and camera disturbances to the radiative field within ice.

Firstly, it is important to minimize disturbance to the incident radiation by avoiding movement between the sun and the site of interest and by reducing sources of shading. Ideally, the site should be as homogeneous as possible over a large area, particularly in terms of snow cover. It is also essential to use an upward-looking radiometer to monitor irradiance throughout the experiment, especially under variable sky conditions such as passing clouds. Secondly, the radiative field within the pack

ice must remain as undisturbed as possible, despite the hole made for the measurement. The use of a diffuser to block unwanted light should be considered. Additionally, the camera and boom should be painted to match the reflectance of the surrounding medium, to avoid absorbing light that would alter the radiative field. Thirdly, measuring light in air above the freeboard level presents several challenges due to the reduced radiance caused by the change in refractive index. A satisfactory solution to this problem has yet to be identified."

- **Potential for AOP Validation:**

  **Is there an opportunity to use coincident AOP measurements to validate the derived IOPs? It would be particularly interesting to compare the modeled AOPs to observations from the High Arctic transmission and albedo sites, if feasible.**

Thank you very much for this proposal. We couldn't agree more. Comparing measured and modeled apparent optical properties would greatly increase the confidence of inferred scattering properties. It would also challenge several assumptions made in radiative transfer models.

Nevertheless, this closure experiment was not directly in line with the main objectives of this article. Our principal goal was to demonstrate the possibility of measuring the light field as it transforms in sea ice from the atmosphere to the water column, using the least intrusive tool possible. We also wanted to develop a reproducible method for using these apparent property profiles to deduce the inherent optical properties of sea ice.

However, we are currently working on another paper that will do just that and enable this validation with reflectance and transmittance. As well as studying the impact of these new measurements on the partitioning of radiative energy within the pack ice.

**Minor Comments and Suggested Edits:**

- **L24: Replace "first" with "top" or "surface."**
  It has been replaced by "top" as suggested.
- **L40: Capitalize "Arctic" (check throughout manuscript).**
  It has been corrected throughout the manuscript.
- **L43–49: Consider connecting more explicitly to physical processes mentioned earlier, such as energy partitioning.**
  Thank you for your suggestion, we agree that this would enable a greater understanding of the vertical energy partitioning within sea ice. As mentioned in a response to a previous comment, this will be addressed in another study.
- **L69: Clarify whether this sentence refers specifically to prior uses of cameras as radiometers.**

Thank you for pointing out this ambiguity. The sentence refers to radiometer which can measure the radiance distribution. It has been changed accordingly.

- **L82: Correct "Artic" to "Arctic" and capitalize "North Pole."**
  Thank you for pointing out this error, it has been corrected.
- **L95–96: Consider specifying "landfast ice" if appropriate.**
  It has been added.
- **L100: Since variability is discussed later, add brief notes on surface and sub-surface variability at each site here.**
  A note has been added about the variability. It goes as follows:
  "At both field locations, The experimenters paid particular attention to site selection in order to ensure the largest possible homogeneous area. No keels or under-ice variability were observed."
- **L102: Fix "depth-resolved".**
  Thank you for pointing out the typo. "Depth-resolved" has been corrected accordingly.
- **Figure 1d: No snow is shown at the Chaleur Bay site. Was snow cleared? Descriptions suggest snow-ice may be present—consider revising the structural model.**
  Since we did not extract ice cores or conduct a snow pit, the composition of the surface layer at the Chaleur Bay site remains uncertain. We believe it likely consisted of a mixture of previously melted and refrozen snow, along with sun transformed ice. Due to this uncertainty, we chose to refer to this layer as the "surface" layer, as shown in the geometric model presented in Fig. 1d. We acknowledge that the nature of this layer may be unclear to readers. To address this, we have added the following clarification at lines 93-94:
  "The surface of the ice was covered with very granular snow, indistinguishable from sun-transformed ice. To underline this ambiguity, this part of the ice is referred as "surface slab"."
- **L120: Indicate the time required to acquire a full profile at a site.**
  Thank you for your suggestion. The following sentence was added in the methods section:
  "Each profile measurement took approximately 10 to 15 minutes, depending on the vertical resolution and ice thickness."
- **L190: Should this be "below" or possibly "drained layer" instead of "skeletal layer"?**
  In this case, we meant the 8 centimetres at the bottom of the ice, just before the interface with the water column. Not 8 centimetres above the freeboard, as you may have understood. The sentence has been rephrased because this is indeed a layer associated with a skeletal layer.
  "The eight remaining centimeters above the ice-seawater interface were considered the skeletal layer."
- **Figure 2: Include RGB color/wavelength labels in the image labels/caption.**
  Thank you for your comment, labels have been added.
- **Figure 3: Add "High Arctic" to the caption. Consider making the "HL simulations" line more visible or matching the legend.**
  Thank you for your comment, the lines have been modified to be more visible.

- **L326: Estimate the percentage of the view impacted by the operator's shadow. At what depth does this become negligible?**
Thank you for your comment, this is indeed an important element for most radiometric measurements, as the operator has an influence on the light field. We have taken great care to minimize this effect. Firstly, we have always made an effort to stay as far away from the measurement hole as possible. Secondly, only strictly necessary personnel were positioned close to the hole. Thirdly, the hole always faced the sun, so that the operator's shadow was behind him. We reasonably believe that the effect of the operator's shadow disappears completely once the camera is fully inserted in the hole and the operator is no longer in the field of view of the camera.

But because we love radiometry, we're going to take the reflection a little further. It's important to distinguish between two different problems. Firstly, the presence of the operator in the camera's field of view, and secondly, the impact of the operator's shadow on the amount of light incident on the surface.

The first problem only applies to the first measurement made at the interface, because only then is the operator visible to the camera. Once the camera is in the middle, it can no longer see the operator. Below is a measurement taken at the surface, where the operator's silhouette is clearly visible. In the upper $2\pi$ steradians, the operator's silhouette represents about one eighth of the field of view. If the illumination is completely diffuse, this would have about the same impact on the amount of light measured by the camera. In the case of direct illumination, this effect is greatly reduced as long as the operator does not directly block the sunlight. Four solutions to this problem seem possible and relatively easy to implement. Firstly, a tripod could be used, and the camera triggered remotely by Wi-Fi, so that as long as it's not in the water, it can receive the signal. Secondly, in diffuse illumination cases, the operator could be dressed in white having the same reflectance as the clouds, thus minimizing its effect on the light field. Thirdly, linear interpolation could be used to fill in the operator's silhouette with measurements. Fourthly, the first surface measurement could be ignored, which is what we have decided to do in this work.

[Figure]

$L_{600nm}(0.0 cm) [W \cdot sr^{-1} \cdot m^{-2} \cdot nm^{-1}]$

$L_{540nm}(0.0 cm) [W \cdot sr^{-1} \cdot m^{-2} \cdot nm^{-1}]$

$L_{480nm}(0.0 cm) [W \cdot sr^{-1} \cdot m^{-2} \cdot nm^{-1}]$

Secondly, the case of the operator's shadow on the pack ice is more complex, as it has an influence on the amount of energy in the medium, and its influence depends on the diffusion coefficient of the pack ice, which is precisely what we're trying to determine. For the sake of simplicity, we'll only consider the case where illumination is direct and consider a case where the ice is highly diffusive and another where the ice is transparent. In the highly scattering case, photons travel in all directions and can cover significant distances in the medium before exiting towards the atmosphere. We therefore believe that the effect of the shadow will be blurred by the significant scattering. In the case of very weak scattering, as might be the case in freshwater ice. It is likely that the operator's shadow will have a very significant effect and will even be visible from beneath the ice. For the impact on the incident radiation, let's imagine an ideal case where we consider that the camera receives the incident signal uniformly over a circle with a radius of 2 meters, and that the operator is a cube of 50 centimeters wide and two meters high. We also assume that the shadow is perfectly black. Less than 10% of the signal will be blocked by the shadow.

[Figure]

In the specific case of this study, the surfaces were highly scattering and the light field was azimuthally homogeneous, which confirms the hypothesis that shadows are not visible in the measurements. If one had to introduce a mathematical criterion for the depth at which the light field becomes homogeneous, we believe we would use the inverse of the reduced scattering coefficient, 1/b'. This corresponds, on average, to the distance a photon must travel to undergo enough scattering events for its direction to become completely random. In the case of the surfaces measured in this study, this distance is approximately 1.5 millimeters for

the High Arctic and 1.4 centimeters for Chaleur Bay. It is nevertheless impossible to precisely determine the effect on the incident energy on the sea ice, but we believe that this effect is relatively small compared to other uncertainties, such as the modeling of sea ice in a single dimension while ignoring horizontal variability.

- **Figure 4: Add labels for "High Arctic" and "Chaleur Bay."**
  Thank you for the suggestion. We have added the labels "High Arctic" and "Chaleur Bay" to the figure, positioned at the top of the central graphic.
- **L370: Correct phrasing to "give rise."**
  It has been corrected, thank you for your comment.
- **L375: Clarify whether "surface slab" refers to snow or a surface scattering layer.**
  Thank you for your comment. As explained in an answer above, we decided to refer to this layer as the "surface layer".
- **L384: Should "sea ice" be "water" here?**
  Yes, you are correct, it has been replaced accordingly.
- **L407: Fix typo: "the in"**
  The typo has been corrected.
- **L407: Consider changing "episodes" to "cycles."**
  This sentence was reformulated.
- **L418: Reword for clarity: the phrase "where contamination sediment sources is far away" is awkward. Perhaps rephrase to emphasize the likely absence of sediment due to MYI origin or absence in cores.**
  This sentence has been rephrased.
- **L425: Replace "wrong" with a more scientific term such as "nonphysical."**
  The sentence has been rephrased, thank you for your comment.
- **L481: Capitalize "High" for consistency with "High Arctic."**
  Thank you for this suggestion. The letter has been capitalized for consistency.
- **L515–517: Could this issue be addressed in future work by increasing the number of measurements within a site?**
  Yes, this text has been added to express this.
  "To characterize the effect of horizontal variability, several measurements with the camera could be taken. For example, it would be interesting to perform a transect with the camera, gradually approaching a melt pond, and observe the distance at which the light field begins to deviate from azimuthal homogeneity."

---

## Author Response (AR2)

**Response to editor**

The author's have adequately addressed all of the referees suggestions and concerns. I recommend publication after some very minor technical corrections, listed below:

We would like to thank the editor.

Line 84 – should this read "three RAMSES-ACCVIS sensors"

Thank you for you comment, it has been changed accordingly.

Line 206 - enhanced

It has been corrected.

Line 248-9 – wrong reference style is used here

Thank you for pointing out this error, it has been corrected.

Line 290 – "loss" not "lost"

Thank you for pointing out this error, it has been corrected.

Line 378 – "These discrepancies are larger"

Thank you, it has been corrected.

Line 429-430 – by "latter", are you referring to the iteration numbers?

Yes we mean the iteration numbers, "latter" was changed for "iteration numbers" to ensure clarity.

Line 412 – sits

Thank you for pointing out this error, it has been corrected.

Line 509 – I think you mean "snow ice" here. i.e. formed from the freezing of water-inundated snow on the surface. Superimposed ice forms from the refreezing of melted snow.

Indeed, we were referring to ice that forms when the freeboard becomes negative and water floods the bottom to the snow layer. It has been changed accordingly.

Line 578 – "we would like to emphasize this new possibility..."

The suggested change was made.

**Line 585 – snow ice**

See above.